# The Role of Post-Translational Acetylation and Deacetylation of Signaling Proteins and Transcription Factors after Cerebral Ischemia: Facts and Hypotheses

**DOI:** 10.3390/ijms22157947

**Published:** 2021-07-26

**Authors:** Svetlana Demyanenko, Svetlana Sharifulina

**Affiliations:** 1Laboratory of Molecular Neurobiology, Academy of Biology and Biotechnology, Southern Federal University, pr. Stachki 194/1, 344090 Rostov-on-Don, Russia; svetlana.sharifulina@gmail.com; 2Neuroscience Center HiLife, University of Helsinki, Haartmaninkatu 8, P.O. Box 63, 00014 Helsinki, Finland

**Keywords:** cerebral ischemia, histone acetylation, histone deacetylases, histone deacetylase inhibitors, non-histone proteins

## Abstract

Histone deacetylase (HDAC) and histone acetyltransferase (HAT) regulate transcription and the most important functions of cells by acetylating/deacetylating histones and non-histone proteins. These proteins are involved in cell survival and death, replication, DNA repair, the cell cycle, and cell responses to stress and aging. HDAC/HAT balance in cells affects gene expression and cell signaling. There are very few studies on the effects of stroke on non-histone protein acetylation/deacetylation in brain cells. HDAC inhibitors have been shown to be effective in protecting the brain from ischemic damage. However, the role of different HDAC isoforms in the survival and death of brain cells after stroke is still controversial. HAT/HDAC activity depends on the acetylation site and the acetylation/deacetylation of the main proteins (c-Myc, E2F1, p53, ERK1/2, Akt) considered in this review, that are involved in the regulation of cell fate decisions. Our review aims to analyze the possible role of the acetylation/deacetylation of transcription factors and signaling proteins involved in the regulation of survival and death in cerebral ischemia.

## 1. Introduction

Acetylation of histones and non-histone proteins modulates gene expression and signaling in cells. Changes in the secondary structure of proteins by acetylation leads to a change in their enzymatic activity, subcellular localization and protein-protein interactions [1]. 

The study of non-histone protein acetylation/deacetylation began after the success of the clinical use of histone deacetylase inhibitors (HDACs) in the treatment of various forms of cancer and was driven by the search for the causes of cytotoxicity of nonselective HDAC inhibitors (HDACi) [2]. Non-histone substrates of HDAC and acetyltransferases (HAT) have been identified, which are tumor suppressors, signaling mediators, steroid receptors and transcription factors [3,4]. The number of identified proteins in which activity is regulated by acetylation/deacetylation to date is certainly lower than the actual amount represented by acetylome in vivo.

There are very few studies on the non-histone protein acetylation/deacetylation in brain cells, and there is apparently no data on these processes in stroke. Our review aims to analyze the possible role of acetylation/deacetylation of transcription factors and signaling proteins involved in the regulation of apoptosis in ischemia.

## 2. Protein Acetylation and Deacetylation Enzymes

Acetylation and deacetylation of histones and non-histone proteins is carried out by histone deacetylase (HDAC) and histone acetyltransferase (HAT). Histone acetyltransferases (HATs) transfer acetyl groups from acetyl coenzyme A (Acetyl Co-A) to the ε-amino group of lysine residues, while histone deacetylases (HDACs), on the contrary, catalyze the removal of acetyl groups. Since histones were the first identified targets of deacetylases and acetyltransferases, these enzymes were named histone deacetylases and histone acetyltransferases. However, in addition to regulating transcription, HAT/HDAC regulate the most important functions of cells by acetylating/deacetylating a huge number of non-histone proteins, which have always been their evolutionarily primary targets [3]. These proteins regulate cell survival and death, replication, DNA repair, the cell cycle, and cell responses to stress and aging.

## 3. Histone Acetyltransferases

Depending on the intracellular localization, HATs are classified into types A or B, which either contain or do not contain a bromodomain [5] (Figure 1). Type A HATs are mainly responsible for acetylation associated with transcription. Type B cytoplasmic HATs acetylate de novo synthesized histones and non-histone proteins. Based on sequence homology as well as common structural features and functions, HATs have been grouped into three main categories: GNAT (GCN5-related N-Acetyltransferases), EP300/CREBBP (E1A binding protein p300/CREB-binding protein), and the MYST family. PCAF (P300/CBP-associated factor), belonging to the GNAT family, is the most important enzyme that acetylates non-histone proteins [6]. In addition, PCAF is the only HAT in which, even with a complete knockout, no phenotypic changes are observed [7]. In the model of photothrombotic stroke (PTS), it was shown that PCAF is poorly expressed in normal neurons and astrocytes of the rat cerebral cortex. However, the protein level increased in neurons and especially in the astrocytes of the penumbra 4–24 h after PTS [8]. Intracellular localization and activity of PCAF is regulated by its acetylation. Autoacetylation of PCAF, or its acetylation by p300, enhances the acetyltransferase activity of the enzyme and leads to its translocation into the nucleus. Deacetylation of PCAF by HDAC3 decreases the activity of the enzyme and promotes its cytoplasmic localization [9].

HAT1 is considered as a cytoplasmic protein. The enzyme acetylates newly synthesized histones in the cytoplasm before being imported into the nucleus [10]. PTS-induced upregulation of HAT1 and PCAF occurred in the penumbra due to HAT1 and PCAF overexpression in the cytoplasm of neurons and astrocytes [8]. This indicates the possibility of acetylation of the cytoplasmic proteins HAT1 and PCAF during cerebral ischemia.

## 4. Histone Deacetylases

Four classes of HDACs are distinguished according to the functions, cell localization, and expression patterns in mammals. Classes I (HDACs-1, -2, -3 and -8), II (HDACs-4, -5, -6, -7, -9 and -10) and IV (HDAC-11) are zinc-dependent enzymes, while in class III enzymes (Sirtuins), nicotinamide adenine dinucleotide (NAD+) acts as a cofactor and HDAC1 and HDAC2 are parts of Sin3, NuRD, CoREST, and NODE complexes that suppress transcription [11] (Figure 2). HDAC1 may play a dual role in the regulation of neuronal life and death. If HDAC1 interacts with HDAC3, it leads to neuronal death, but it is neuroprotective when it interacts with HDRP, a shorter form of HDAC9 [12]. HDAC3 is part of the NCoR/SMRT corepressor complex and regulates gene expression by the deacetylation of histones, as well as a number of non-histone proteins [13]. Once cerebral ischemia has occurred, HDAC1 is able to enter the cell cytoplasm [14]. In contrast to HDAC1 and HDAC3, HDAC2 expression is increased not only in the nuclei of neurons, but also in the nuclei of penumbral astrocytes on the first day after PTS and during the recovery period [15]. It has been shown that an increase in HDAC2 expression plays a crucial role in the survival or death of neurons in the peri-infarction region of the cerebral cortex of animals, both in the PTS model and after middle cerebral artery occlusion (MCAO) [16,17]. Inhibition of HDAC2 promoted the restoration of brain function, while overexpression increased stroke-induced functional impairment. At the same time, inhibition of other HDAC isoforms was ineffective ([17,18]. A recent study by Shoyaib et al. has shown no substantial effect of Panobinostat (pan-HDACs inhibitors) or Entinostat (inhibitors of HDAC1/HDAC3) on motor recovery in mice after photothrombotic stroke. This was accompanied by negligible changes of parvalbumin-positive neurons and comparable infarct volumes among experimental groups, while a dose-dependent increase in acetylated histone 3 was observed in the peri-infarct cortex of drug-treated animals [19].

Another class I histone deacetylase HDAC8 is present mainly in the cytoplasm of neurons and astrocytes of the cerebral cortex, amygdala, hippocampus, and hypothalamus [15,20]. HDAC8 expression in neurons and astrocytes of the cerebral cortex of mice was significantly increased during the recovery period, from 3 to 14 days after PTS [15].

Overexpression of HDAC3, HDAC6, and HDAC11 was observed in the penumbra at 3 and 24 h after MCAO and persisted for a week after reperfusion. Inhibition of HDAC3 or HDAC6 expression increased cell viability [21]. After PTS, HDAC6 expression was high both on the first day, and in the early recovery period [22]. Decreased HDAC6 activity caused by the selective HDAC6 inhibitors tubastatin A [23] or HPOB [22], restores acetylation of α-tubulin, a classic substrate of HDAC6, and reduces apoptosis of nerve cells, thereby protecting the brain tissue from damage. HDAC6 is a predominantly cytoplasmic enzyme. However, HDAC6 localization in cells is regulated by nuclear import and export [24]. HDAC6 could have nuclear localization and interoperate with HDAC11 [25], p300 [26], repressor complex LCoR and p65 subunit of nuclear factor-κB [27].

It is known that class II HDACs recruit class I HDACs to form the NCoR/SMRT complex, thereby suppressing the transcription of a number of proteins [28]. Data on the role of HDAC4 in neurodegeneration and neuroprotection are contradictory. Some authors reported the ability of HDAC4 to support neuronal survival [29,30]. Others did not find a relationship between neuronal survival and HDAC4 expression [22,31]. HDAC4 rapidly translocates into the nucleus in response to a decrease in potassium or an increase in glutamate in cultured neurons, which induces neuronal cell death [32,33]. Kassis et al. reported that HDAC4 nuclear localization promotes brain recovery after stroke [34]. A decrease in HDAC4 expression and its translocation into neuronal nuclei was noted both on the first day, and 2 weeks after stroke [22,35]. It has been shown that miR-29a-3p strengthened the effect of dexmedetomidine on improving neurologic damage in newborn rats with hypoxic-ischemic brain damage by inhibiting HDAC4 [36].

Another representative of the HDAC IIa class, HDAC5, is involved in the differentiation of neurons, and regulates the survival of neurons in the cerebral cortex through the action of factors that cause apoptosis [37]. Overexpression of HDAC5 and its nuclear localization caused apoptosis of cultured neurons of the granular layer of the cerebellum [38]. On the other hand, the nuclear export of HDAC5 stimulates the regeneration of sensory neuron axons after injury [39]. HDAC4 and HDAC5 physically interact with the transcription factor downstream regulatory element antagonist modulator (DREAM). A recent study showed that the DREAM/HDAC4/HDAC5 complex epigenetically down-regulates ncx3 gene transcription after stroke, and the pharmacological inhibition of class IIa HDACs reduces stroke-induced neurodetrimental effects [40].

Sirtuins are class III histone deacetylases. Seven sirtuins, 1–7 have been identified in mammals. Sirt1 and Sirt6 are mainly localized in the cell nucleus, Sirt7 in the nucleoli, Sirt3, Sirt4 and Sirt5 are mitochondrial proteins, and Sirt2 is located in the cytoplasm. Sirt1 and Sirt2 are the most studied. There is a significant amount of data indicating the neuroprotective properties of Sirt1 in ischemic stroke, brain injury, and neurodegenerative diseases [41,42,43]. Sirt1 knockout mice displayed larger infarct volumes after ischemia than their wild-type counterparts in the MCAO model [44], while mice with Sirt1 overexpression were more resistant to ischemia [45]. Sirt1 activators reduce the size of the infarction [46]. In contrast, Sirt2 is usually assigned a proapoptotic role, and pharmacological inhibition or knockdown of Sirt2 can prevent neuronal apoptosis in ischemic stroke [41,47,48]. Our results show that during the recovery period after PTS, an increase in Sirt1 and Sirt2 is observed, but with opposite functional consequences. Nevertheless, several studies indicate that the function of Sirt2 in the ischemic brain is much more complex than simply pathological or neuroprotective and depends on the cellular and intracellular localization of different enzyme isoforms, the level of its phosphorylation, and the type of substrate [41,42,48]. It has been shown that acute cerebral ischemia-induced downregulation of Sirt3 protein expression contributes to neuronal injury via damaging mitochondrial function [49]. Sirt3 plays a protective role in ischemic stroke via regulating HIF-1α/VEGF signaling in astrocytes [50]. Moreover, SIRT6 exerts a protective role in ischemic stroke by blunting I/R-mediated damage to the blood-brain barrier [51].

## 5. Post-Translational Modifications of HDACs

Post-translational modifications of HDACs are capable of affecting their deacetylase activity. Thus, phosphorylation of HDAC2 at serine S394, S422, and S424 activates the enzyme [52], while S-nitrosylation of cysteine (C262 and C274), on the contrary, inhibits the enzyme in muscular dystrophy in mice [53]. In a model of cardiac hypertrophy in mice, it was shown that in the nuclei of cardiomyocytes, PCAF interacts with HDAC2 and acetylates the protein at lysine 75. This leads to the phosphorylation of the protein (S394) and its activation, and HDAC5 moves from the cytoplasm of cells to the nucleus, where it deacetylates HDAC2, reducing the hypertrophy of cardiomyocytes [54]. This illustrates the two opposing actions of prohypertrophic HDAC class I and antihypertrophic HDAC class IIa, which should be considered when developing new HDAC inhibitors, since HDAC IIa inhibitors can aggravate the disease by acetylating and thereby activating other HDACs.

## 6. Biological Activity of HDAC Inhibitors

HDAC inhibitors (iHDACs), which were effective in protecting the brain from ischemic damage, belonged to two chemical groups: (a) small carboxylates: valproic acid (VPA), sodium butyrate (SB), and sodium 4-phenylbutyrate (4-PBA); (b) Hydroxamate-containing compounds: suberoylanilide hydroxamic acid (SAHA) and trichostatin A (TSA) and others [55]. In cellular and animal models of ischemia, iHDACs have been shown to protect the brain from excitotoxicity, oxidative stress, endoplasmic reticulum stress, apoptosis, inflammation, and BBB damage [40,56,57]. They also promote angiogenesis, neurogenesis, and stem cell migration to damaged areas, which leads to functional recovery after brain ischemia [58,59]. However, classic iHDACs are not selective. They inhibit HDACs in classes I or II, or both. The use of nonselective iHDACs for cancer treatment in humans has caused side effects ranging from minor (e.g., diarrhea, anorexia, dehydration) to severe (e.g., myelosuppression, thrombocytopenia, and cardiotoxicity) [60,61,62]. Because iHDACs were originally developed to treat a variety of cancers, they can treat chronic neurodegeneration and promote recovery from stroke without the fear of increasing the probability of cancer developing. However, the use of nonselective iHDACs for the treatment of neurodegenerative diseases or the repercussions of stroke can have a number of side effects. VPA and TSA, nonselective iHDACs, promote cell cycle arrest by preventing the formation of mature oligodendrocytes [63,64] and have a cytotoxic effect on cultured cerebellar and cortical neuronal cells [65,66,67]. In addition, the increase of double-stranded DNA breaks and apoptosis were shown in cortical neurons when only nonselective iHDAC was administered [67] or in combination with DNA damaging agents [68]. Nonselective iHDACs cause the death of dopaminergic neurons and neurons in the ventral midbrain [69,70]. SAHA negatively affects the survival of oligodendrocyte progenitor cells and prevents their differentiation into mature oligodendrocytes, which can slow down axonal myelination during brain repair after injury, as well as in the treatment of mental and neurodegenerative conditions [71]. The HDAC3 inhibitor RGFP966 ameliorated ischemic brain damage by downregulating the AIM2 inflammasome [72]. Brain penetrant benzazepine-based HDAC6 inhibitors reduced cerebral infarction and alleviated neurobehavioral deficits in post-ischemic treatment in rats with transient middle cerebral artery occlusion (MCAO) [73].

Although HATs, like HDACs, are involved in tumor progression and their inhibitors may be useful in the treatment of cancer, they are not currently used in clinical practice. Due to their ability to be a part of protein complexes, modern HAT inhibitors are unstable, have low activity, or lack of selectivity [74]; however, the development of more selective HAT inhibitors will undoubtedly be of interest for future cancer therapy [75], and possibly for stroke therapy.

The active search for non-histone HAT/HDAC targets is due to the search for the causes of the cytotoxicity of nonselective iHDACs used in the treatment of various forms of cancer. Changes in the acetylation pattern of the transcription factors or signaling proteins that regulate cell survival or death by nonselective iHDACs can reactivate or inhibit them, leading to a change in cell fate. Besides, the use of nonselective iHDACs may cause dysfunction of chaperones and the inhibition of pathways that regulate stress responses in the endoplasmic reticulum [76,77].

Thus, HDACs are widely represented in the brain. Some of them are located in the nuclei of brain cells, some in the cytoplasm, and others in both the nucleus and the cytoplasm. Their functions are different. The role of different HDAC isoforms in the survival and death of brain cells after stroke is controversial. Some HDACs mediate survival processes, while others are involved in neurotoxic reactions of cells after ischemic stroke. Some of them can exhibit both neuroprotective and pathological properties depending on the type of cell, their intracellular localization, and the nature of post-translational enzyme modifications. Expression of HDAC1, HDAC2, HDAC3, HDAC4, HDAC6, Sirt1, and Sirt2 is increased in the brain after ischemia. It is not known whether acetylation/deacetylation of different HDACs occurs in brain cells and what their functional consequences are. Further studies of changes in the dynamics and nature of HDAC interactions with each other, with cytoplasmic proteins and with proteins of repressor complexes, are required. 

Thus, there is a difference in the effects of HDAC inhibitors. They are effective in the treatment of several types of cancer; promoting cancer cell apoptosis. On the other hand, the neuroprotective effect of inhibitors is beyond doubt. To elucidate the reasons for such differences, it is necessary to study the effect of reversible acetylation on the most important transcription factors, the regulatory proteins involved in the survival or death of nerve cells after ischemia.

## 7. Non-Histone Substrates of HAT and HDAC

Non-histone HAT/HDAC substrates include tumor suppressor proteins (e.g., p53, RUNX3), signaling mediators (e.g., STAT3, β-catenin, SMAD7), steroid receptors (e.g., androgens, estrogen, SHP), transcription factors, and coregulators (for example, c-Myc, HMG, YY1, EKLF, E2F1, GATA factors, HIF-1α, MyoD, NF-κB, FoxB3), as well as structural (for example, cell motility proteins), chaperone, and nuclear import proteins (e.g., α-tubulin, importin-α, Ku70, HSP90) [3,4,78]. Acetylation of non-histone proteins can affect many molecular functions of these proteins, such as mRNA splicing, mRNA transport and integrity, protein translation, protein activity, localization, stability and interactions [3,79,80], and this list is updated every year. These proteins determine the growth, differentiation, migration, and survival of cells, both in normal conditions and when damaged. Therefore, acetylation-dependent signaling pathways are key determinants of homeostasis.

Proteomic studies of the expression of hundreds of proteins in the penumbra after PTS indicate a consistent increase in the level of many signaling proteins that can initiate, mediate, or regulate apoptosis, as well as a number of proteins with an antiapoptotic effect [81,82]. The development of apoptosis was indicated by the increased expression of proapoptotic proteins like p53, p38, p75, c-Myc, E2F1, JNK, AIF, Par4, DYRK1A, NMDAR2a, GADD153, GAD65/67, Smac/DIABLO, caspases, and PSR, and a decrease in the level of Hsp70. However, at the same time, the level of antiapoptotic proteins increased, including receptors for growth factors EGFR and estrogens, protein kinases ERK 1 and 5, Akt, phosphatase MKP-1, proteins p63, p21Waf-1, and MDM2. Many of these proteins are acetylated and deacetylated. 

The main proteins that play a central role in coordinating cell fate decisions are considered in this review in more detail.

## 8. c-Myc

One of the main regulators of many target genes is the transcription factor c-Myc. It activates (or sometimes suppresses) 10–15% of all genes involved in the regulation of energy metabolism, protein synthesis, oncogenesis, the cell cycle, and apoptosis. It functions at both the transcriptional and epigenetic level. It can potentiate apoptosis. c-Myc is an oncogene in many human tumors [83]. c-Myc regulation can be involved in several signaling pathways, such as JAK/STAT, Wnt/β-catenin, Notch, and the Ras/PI3K/AKT/GSK-3 signaling pathways, that increase c-Myc levels [84].

Its overexpression was also noted after transient global or focal cerebral ischemia in rodents, where c-Myc promoted neuronal death [85]. An increase in the level of c-Myc was observed in the penumbra on the first day after PTS [81].

The stability of c-Myc in different types of cancer cells is associated with its acetylation at lysine 323 by PCAF acetyltransferase (Figure 3) [86]. In lymphoma cells, SIRT1 interacted with the C-terminus of c-Myc and deacetylated it both in vitro and in vivo [87]. However, inhibitors of HDAC, but not sirtuins, increased the acetylation of c-Myc at lysine 323 and inhibited tumorigenesis [88], which promoted the association of c-Myc with Max, a partner required for c-Myc activation. HDAC inhibitors downregulated c-Myc by blocking GSK-3 phosphorylation and exhibited synergistic cytotoxic and c-Myc-suppressive effects (Figure 4) [89]. HDAC3 also deacetylated c-Myc at lysine 323 in cholangiocarcinoma cells, which protected the protein from ubiquitin-dependent proteolysis [90]. Thus, it can be assumed that at least one of the HDACs in brain cells is c-Myc deacetylase. c-Myc stimulates the expression of p53 and E2F1 [91].

## 9. E2F1 

E2F1 transcription factor is one of the key players in determining the fate of the cell. It controls the expression of various genes that regulate DNA synthesis and repair, the cell cycle, and apoptosis [92,93]. E2F1 stimulates apoptosis when the cell cycle is disrupted or suppressed, which is typical of neurons [94]. Its synthesis is controlled by the p38 MAP kinase and the c-Myc transcription factor [95]. E2F1 induces the expression of various proapoptotic proteins, such as caspases 3, 7, 8, and 9, Smac/DIABLO, Apaf-1, Bcl-2, p53, and p73 [94,96]. Overexpression of E2F1, p53, c-Myc, p38, Smac/DIABLO, Bcl-x, caspases 3, 6, and 7 was observed in the penumbra on the first day after PTS [81]. This is consistent with data on the increased expression of E2F1 [97] and p53 [98] in the penumbra after MCAO and in the axotomized spinal ganglia of rats [99]. Inhibition of the E2F1/p53 pathway prevents neuronal apoptosis [96].

iHDACs have been shown to affect E2F1 activity [100]. In cancer cells, in response to genotoxic stress caused by doxorubicin, E2F1 is acetylated by PCAF at three lysines (K117, 120, and 125). This stabilizes the protein and increases its specific binding to DNA [101,102]. Acetylation of these lysines induces the accumulation of ubiquitinated but stable E2F1 [103]. Acetylation of E2F1 promotes the recruitment of chromatin-modifying enzymes and DNA double-strand break repair factors [104].

HDAC1 acts as a deacetylase in different types of cancer cells [101,105,106]. In retinal epithelial cells, E2F1 is deacetylated by Sirt1, which contributes to the resistance of cells to oxidative stress caused by H2O2 [107]. Thus, acetylation/deacetylation of E2F1 can contribute to the resistance of different types of cells to damage. However, in the literature available to us, we failed to find information on the acetylation/deacetylation of E2F1 in brain cells, either in normal conditions or in pathology.

## 10. p53 

Protein p53 is the most studied non-histone HAT and HDAC substrate. p53 is a known promoter of apoptosis. It controls the transcription of hundreds of genes involved in the regulation of DNA repair, cell cycle arrest, metabolism, mRNA translation, apoptosis, and autophagy [108]. The negative regulators of p53 are p21WAF-1, p67, and MDM2 [82]. The role of p53 acetylation/deacetylation in the regulation of gene expression and intracellular signaling pathways is extremely important. p53 was the first non-histone protein to be discovered with activity that was dependent on acetylation. In different types of cancer cells, the C-terminal lysines in p53 are acetylated by p300/CBP (lysines 373 and K382) and PCAF (lysine 320) both in vitro and in vivo [109]. Acetylation of p53 significantly enhances its activity in response to DNA damage. In normal cells, unacetylated p53 can activate genes that are involved in its downregulation, for example, Mdm2. Upon DNA damage, the acetylation of p53 disrupts the interaction between Mdm2 and p53 and recruits HAT to the promoters of genes involved in DNA repair and cell cycle control. Acetylation of p53 causes the activation of proapoptotic genes.

In different types of oncotransformed cells, it has been shown that HDAC1, HDAC2, HDAC3, HDAC6, HDAC8, and SIRT1 can deacetylate p53, which leads to a decrease in protein activity and the repression of transcription [110,111,112]. In colorectal cancer cells, HDAC6 deacetylates p53 at lysines 381/382. The C-terminal lysine residues within p53 also deacetylate HDAC1, HDAC2, and SIRT1. Romidepsin, a specific inhibitor of HDAC1/2, enhances p53 acetylation at lysines 320/372, but not lysines 381/382. SIRT1 preferentially deacetylates p53 at lysine 382 [113]. A decrease in the level of acetylated p53 in patients with colorectal cancer is associated with an increase in HDAC6 expression. The HDAC6 inhibitor A452 reduces the amount of nuclear HDAC6 and therefore the interaction between HDAC6 and p53, which leads to an increase in p53 acetylation at lysines 381/382.

Sumoylation of HDAC2 at lysine 462 in colorectal carcinoma cells allows HDAC2 to bind to p53. Deacetylation of p53 at lysine 320 by sumoylated HDAC2 blocks p53-dependent expression of genes for cell cycle control and apoptosis, reducing apoptosis caused by DNA damage [114]. Genotoxic stress induces desumoylation of HDAC2, which activates p53 and stimulates apoptosis. In lymphoma cells, p53 interacts with HDAC1, HDAC3, and HDAC8 and becomes deacetylated, which reduces their apoptosis [110,115]. Autophagic feedback-mediated degradation of one of the catalytic subunits of the IκB kinase requires p300/CBP-dependent acetylation of p53 during arsenite-induced proapoptotic responses in human hepatoma cells [55]. 

In a rat model of hemorrhagic stroke, it was shown that SIRT1 can reduce neuronal apoptosis and cerebral edema by deacetylating p53 [116]. Thus, acetylation/deacetylation regulates p53 activity by altering the expression of target genes. Moreover, in different types of cells, different isoforms of HAT and HDAC are involved in protein acetylation/deacetylation.

## 11. ERK1/2

It is known that some of the non-histone substrates of HDAC6 are the protein kinases ERK1 and ERK2, which provide the resistance of nervous tissue to ischemic damage [82]. ERK1 can phosphorylate HDAC6, thereby increasing HDAC6-mediated tubulin deacetylation. In turn, ERK1/2 are acetylated by PCAF at lysine 72, which reduces the enzyme activity towards the transcription factor ELK1. This is a well-known substrate of ERK1, and is deacetylated by HDAC6, which promotes ERK1 activation and prevents cancer cell apoptosis [117]. It has been shown that an inhibition of HDAC1 and HDAC6 downregulated the expression of phospho-ERK1 in human head and neck squamous cell carcinoma cells [118]. 

## 12. Akt

Another non-histone substrate of HDAC6 with antiapoptotic activity is protein kinase Bα (Akt). In response to brain ischemia, an increase in Akt expression is observed in penumbral cells [81,119,120]. In human neuronal progenitor cells, Akt is acetylated at lysines 163 and 377 in the kinase domain of the enzyme [121]. The inhibition of HDAC6 can lead to a decrease in the ability of Akt to bind PIP3, which is located in the plasma membrane. This is accompanied by a decrease in the ability of Akt to phosphorylate downstream targets, even when the protein is phosphorylated at serine 473, which usually enhances the catalytic activity of the enzyme [122]. It has been shown that HDAC inhibition by sodium butyrate did not influence the expression of Akt and phospho-Akt in the brain after hypoxic-ischemic brain injury in rats [123].

SIRT1 can also deacetylate and thus activate Akt and the Akt/GSK3 signaling pathway, which promotes axon growth and formation in embryonic hippocampal neurons [124]. In myocardial cells, Sirt1 and Sirt2 are involved in the deacetylation of Akt at lysines 14 and 20, which leads to the activation of the enzyme [125].

In cancer cells, Akt is acetylated by histone acetyltransferases p300 and PCAF at lysines 14 and 20 [126]. HDAC3 binds to Akt and deacetylates it at lysine 20, promoting protein phosphorylation. Exposure to chemotherapy drugs enhances the interaction between HDAC3 and Akt, which leads to a decrease in Akt acetylation, an increase in its phosphorylation, and a decrease in the sensitivity of leukemic cells to apoptosis [127]. In contrast, HDAC3 inhibitors increase the sensitivity of cancer cells to apoptosis after chemotherapy caused by chemotoxicity. HDAC8 is also able to interact with Akt, reducing its acetylation at lysine 426 in breast cancer cells [128].

Cancer studies indicate that the acetylation of anti- and proapoptotic proteins promote apoptosis, and their deacetylation promotes survival and proliferation. Therefore, HDAC inhibitors have therapeutic potential in cancer treatment [129]. 

## 13. Conclusions and Outlook

The transcription factors and signaling proteins that play an important role in brain cell responses to ischemia undergo acetylation/deacetylation. In different cell types, the acetylation/deacetylation of different regions of non-histone proteins containing lysines occurs and HAT/HDAC activity depends on the acetylation site. Moreover, the activity of the HDACs themselves can be regulated by their acetylation/deacetylation. Depending on the acetylation site localization in structural and functional regions of the protein, reversible acetylation can change the activity of the protein, the substrate binding, the intracellular localization, etc.; therefore, the cell response to ischemia depends on the site of the acetylation.

The vast majority of data on non-histone protein acetylation/deacetylation have been obtained in cancer cells. Data on the effects of stroke on non-histone protein acetylation/deacetylation in brain cells are extremely limited and practically absent. This is an area for further research.

The search for HAT and HDAC isoforms that are capable of acetylating/deacetylating the signaling proteins and transcription factors that regulate apoptosis and other essential functions of brain cells after stroke will contribute to the development of effective neuroprotective therapies based on selective iHDACs. It will also contribute to the search for new substrates and tissue-specific HDAC and HAT inhibitors or activators for treating the consequences of stroke at different periods of the disease.

## Figures and Tables

**Figure 1 ijms-22-07947-f001:**
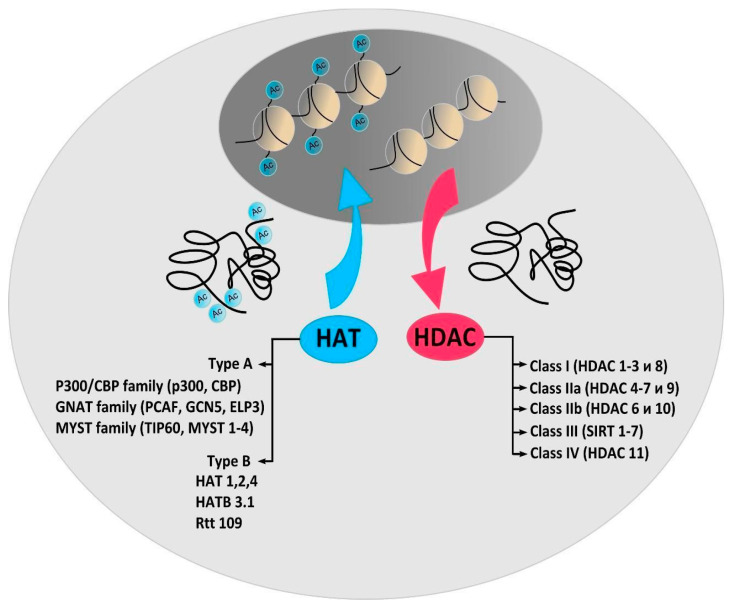
Histone acetyltransferase (HAT) and histone deacetylase (HDAC) classification. The chromatin conformation in the cell according to the HAT/HDAC balance. The different families and classes of enzymes are noted. Type A HATs responsible for acetylation associated with transcription. Type B cytoplasmic HATs acetylate de novo synthesized histones and non-histone proteins. Ac = Acetyl; CBP = cyclic adenosine monophosphate response element-binding (CREB) protein; GNAT = Gcn5-related N-acetyltransferases; PCAF = p300/CBP-associated factor; GCN5 = general control of amino acid synthesis protein 5-like 2; ELP3 = elongation protein 3; MYST = MOZ/YBF2/SAS2/TIP60; TIP60 = TAT interacting proteins 60; SIRT 1–7 = sirtuins.

**Figure 2 ijms-22-07947-f002:**
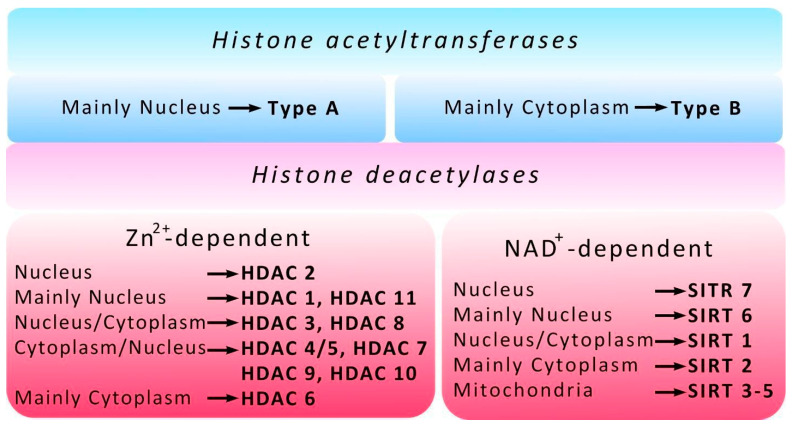
Histone acetyltransferase (HAT) and histone deacetylase (HDAC) localization. The different localizations of HATs/HDACs are shown. Type A HATs are located mainly in the nucleus, Type B HATs are located mainly in cytoplasm. HDAC isoform distribution is shown according to their cell localization and expression patterns in mammalian cells. HDACs -1–11 are zinc-dependent enzymes, Sirtuins (SIRT 1–7) are nicotinamide adenine dinucleotide (NAD+) dependent enzymes.

**Figure 3 ijms-22-07947-f003:**
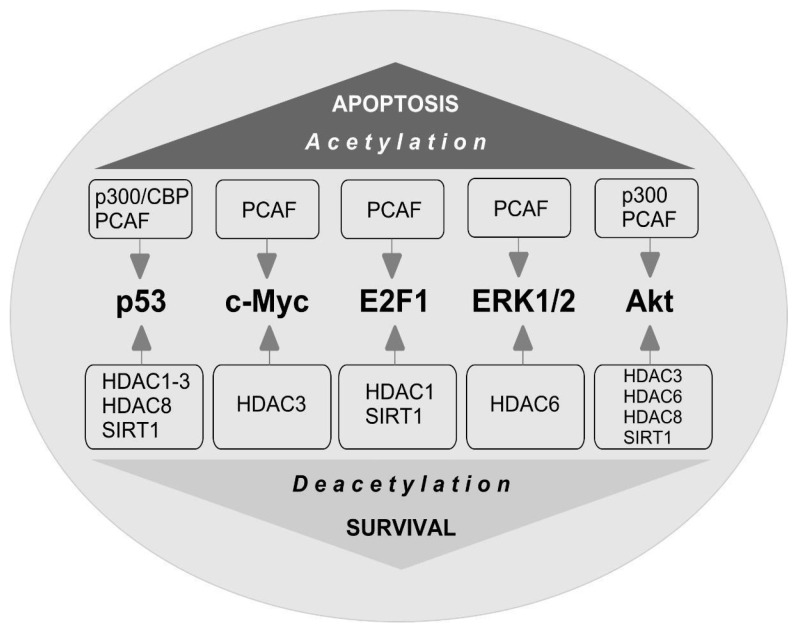
Proteins coordinating the cell fate decision under acetylation/deacetylation conditions. The main proteins that play a central role in coordinating cell fate decisions are shown. p53 is acetylated by cyclic adenosine monophosphate response element-binding (CREB) protein (p300/CBP) and p300/CBP-associated factor (PCAF) acetyltransferases. Acetylation of p53 causes the activation of proapoptotic genes. Histone deacetylases HDAC1, HDAC2, HDAC3, HDAC8, and Sirtuin 1 (SIRT1) can deacetylate p53, which leads to a decrease in protein activity and repression of transcription. c-Myc is acetylated by PCAF. HDAC3 deacetylates c-Myc. c-Myc stimulates the expression of p53 and E2F1. E2F1 is acetylated by PCAF that increases protein specific binding to DNA. E2F1 is deacetylated by SIRT1 and HDAC1. Acetylation/deacetylation of E2F1 can contribute to the resistance of different types of cells to damage. ERK1/2-extracellular signal-regulated kinases are acetylated by PCAF and deacetylated by HDAC6 which prevents cell apoptosis. Akt-protein kinase Bα is acetylated by p300 and PCAF. Akt is deacetylated by HDAC-3,-6,-8 and SIRT1.

**Figure 4 ijms-22-07947-f004:**
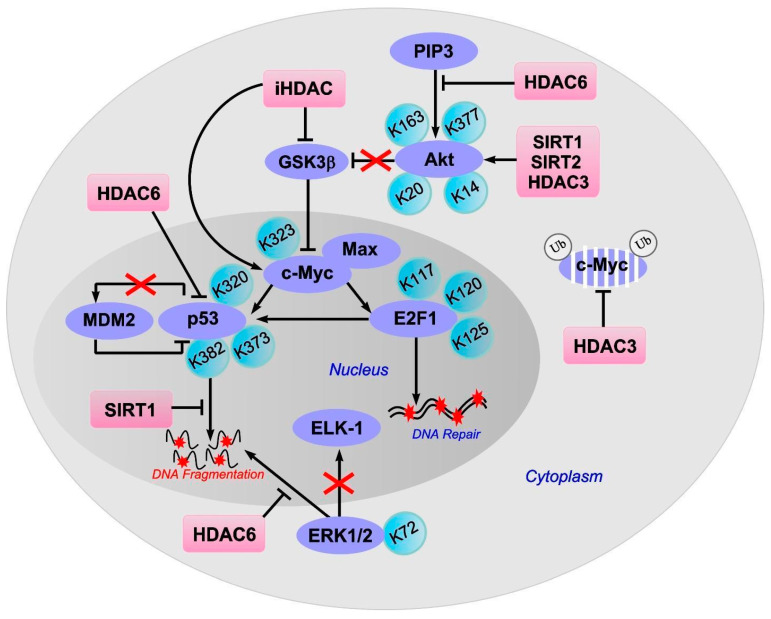
Effect of acetylation/deacetylation on proteins involved in the regulation of apoptosis: c-Myc, p53, E2F1, ERK1/2 and Akt. Inhibitors (iHDAC), but not sirtuins, increase the acetylation of c-Myc at lysine 323 and inhibit tumorigenesis which promotes the association of c-Myc with Max, a partner required for c-Myc activation. HDAC inhibitors downregulated c-Myc by blocking GSK-3 phosphorylation. HDAC3 deacetylates c-Myc at lysine 323, protecting the protein from ubiquitin-dependent proteolysis. c-Myc stimulates the expression of p53 and E2F1. Inhibition of the E2F1/p53 pathway prevents neuronal apoptosis. E2F1 acetylated at three lysines (K117, 120, and 125) promotes DNA double-strand break repair. The C-terminal lysines (K320, K373, K382) in p53 acetylation increases p53 activity in response to DNA damage. Upon DNA damage, the acetylation of p53 disrupts the interaction between Mdm2 and p53 and causes the activation of proapoptotic genes. Sirtuin 1 (SIRT1) can reduce neuronal apoptosis and cerebral edema by deacetylating p53. The HDAC6 inhibition reduces the amount of nuclear HDAC6 and therefore the interaction between HDAC6 and p53, which leads to an increase in p53 acetylation. ERK1/2 acetylation at lysine 72 (K72) reduces the enzyme activity towards transcription factor ELK1, a substrate of ERK1; ERK1/2 deacetylation by HDAC6 promotes ERK1 activation and prevents cell apoptosis. Protein kinase Bα (Akt) is acetylated at lysines 163 and 377 (K163, K377). The inhibition of HDAC6 leads to a decrease in the ability of Akt to bind PIP3. SIRT1 can also deacetylate and thus activate Akt and the Akt/GSK3 signaling pathway. HDAC3 binds to Akt and deacetylates it at lysine 20, promoting protein phosphorylation. Sirtuin1 (SIRT1) and Sirtuin2 (SIRT2) are involved in the deacetylation of Akt at lysines 14 and 20, which leads to the activation of the enzyme.

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
