# Peer review of "The Role of Post-Translational Acetylation and Deacetylation of Signaling Proteins and Transcription Factors after Cerebral Ischemia: Facts and Hypotheses"

_ijms, 2021, doi:10.3390/ijms22157947_

Round 1

Reviewer 1 Report

It is an interesting review showing the most recent studies investigating the effect of acetylation/deacetylation of proteins involved in cell signaling. Overall, the manuscript is well-written, the ideas are exposed clearly, and the images/figures are self-explicative. Reading the paper I found some typos and expressions that were not so clear, such as:

  • In the abstract, “functions of cells by acetylating/deacetylating of histones and non-histone proteins”, the “of” should be omitted.
  • In the abstract, I did not understand the sentence “HDAC/HAT balance in cell effect on gene expression and cell”. 
  • Also in the abstract, on lines 4-6 the information was redundant.
  • On introduction, page 1, I suppose “acetylom” means actually “acetylome”.
  • On page 5, when the authors describe the data of reference 44, it is written “myocardial infarction”, but the paper is actually done in the brain.
  • On discussion, page 11, there is an extra period in the sentence “after chemotherapy.caused by chemotoxicity”, that must be omitted.
  • It is not clear what the authors meant with “while the deacetylation - survival and proliferation” at the end of the discussion.

Author Response

Thank you a lot for your comments and suggestions. They are extremely useful! Please see the attachment with the author's notes. Thank you again for your time and such a detailed review.

Reviewer 2 Report

The review "the role of post-translational acetylation and deacetylation of signaling proteins and transcription factor after cerebral ischemia. Facts and hypotheses”” describes HATs and HDACs functions on no-histones proteins. This topic is interesting but this review is more focused in HAT/HDAC role in cancer than in cerebral ischemia, seeming a review about cancer. In the tittle authors use the words “facts and hypotesis” but the number of facts is really low and almost there are not any hypothese. This reviewer understand the few bibliography of this topic but in my opinion authors may discuss better the posible link between cancer results and cerebral ischemia.and try to focus better the attention on ischemic damage and not in oncology.

Indeed,  a table with HDAc inhibitors effects in brain would be useful and informative.

Finally, in the third paragraph of c-Myc section authors describe acetylation of p53, c-Myc, E2F1, ERK1/2 and Akt. This paragraph should be a introduction about non-histone substrates and should no be included in c-myc section. The information of proteins of apoptosis pathways is messy and is not easy to read, mixing the differnet proteins and goig from c-Myc to E2F1, returning to c-Myc…

In summary, this review should be improved focussing better the attention on ischemic damage and connecting the results in cancer research with the posible applications on ischemia

Author Response

Thank you for your comments and suggestions. Hope the notes explained the importance of the contribution of the review "The role of post-translational acetylation and deacetylation of signaling proteins and transcription factor after cerebral ischemia. Facts and hypotheses” to the field. Please see the attachment. 

Round 2

Reviewer 2 Report

Unfortunately, this reviewer still has some concerns about this manuscript:

  1. It is true that bibliography about acetylation on brain cells under stroke is limited. However, in my opinion this article describes deeply what happens in cancer which is a very different disorder (although both share some common mechanisms). Authors could include bibliography and results about these pathways in other brain system disorders. Indeed, the tittle says facts and hypothesis and authors could give some hypothesis and try to explain better in the manuscript that some of these pahtways (survival, apoptosis…) are important both in cerebral ischemia and cancer. The manuscript should explain better why authors include so many information about cancer acetylation and how this could be extrapolated to brain ischemia.
  2. I agree with authors and the role of HDAC isoforms in survival and death is controversial. However, authors says in yhe text that HDACi are effective protecting the brain from ischemic damage. In my opinion a table in Biological activity of HDAC inhibitors may be useful. This could be a simple table about which researchs have beeb done with different inhibitors (VPA, TSA, SAHA… in brain ischemia. Authors explain this in the text and a table would help the reading of the text.
  3. I still think that the section of non-histone substrates of HAT and HDAC is confuse. Authors describes c-myc acethyltaion in the same section where authors describes p53 or E2F acetylation and after, they explain deepler p53 or E2F acetylation. This section is quite mixed and messy, which makes it more difficult to read. Maybe adding sections and subsections in the text it would be easier.

Author Response

Thank you very much for your comments and suggestions. Please see the attachment with  author's responses. 
